# Farm Household Vulnerability Due to Land and Forest Fire in Peatland Areas in South Sumatra

**Muhammad Yazid** * 🆔, **Dessy Adriani** 🆔, **Riswani and Dini Damayanthy**

Department of Agribusiness, Faculty of Agriculture, Universitas Sriwijaya, Inderalaya 30662, Indonesia; dessyadriani@fp.unsri.ac.id (D.A.); riswani@fp.unsri.ac.id (R.); damayanthy@fp.unsri.ac.id (D.D.)
* Correspondence: yazid_ppmal@yahoo.com

**Abstract:** Land and forest fires in peatland areas in Indonesia have a widespread negative impact on surrounding communities. Possible vulnerabilities relate to economic, social, ecological, livelihoods, and environmental vulnerability. This study aimed to assess household vulnerability due to land and forest fires in peatland areas in Ogan Komering Ilir District in South Sumatra and observe changes in peat ecosystems in those areas. The study was conducted in three peatland hydrological units (PHUs)—(1) PHU Sungai Sugihan–Sungai Lumpur; (2) PHU Sungai Sibumbung–Sungai Batok; and (3) PHU Sungai Saleh–Sungai Sugihan—covering 300 households as samples. Primary data were collected through structured interviews and analyzed descriptively. The analysis revealed the following: (1) PHU Sungai Sibumbung–Sungai Batok had the highest score for livelihood vulnerability and climate change but the lowest score for social, economic, and ecological vulnerability; (2) PHU Sungai Saleh–Sungai Sugihan had the highest score for economic and ecological vulnerability but the lowest score for livelihood vulnerability; (3) PHU Sungai Sugihan–Sungai Lumpur had the highest score for social vulnerability but lowest score for climate change vulnerability; and (4) the number of household members, toddlers, and elderly, and all economic indicators except land ownership, contributed relatively similarly to social vulnerability in all PHUs.

**Keywords:** ecosystem; social; economic; livelihood; ecological; climate change





## 1. Introduction

Peatland is a unique ecosystem in terms of structure and function, with high vulnerability to disturbance [1–4]. Currently, most of the peatland and forests in Indonesia experience severe damage as a result of human activities that pay little attention to environmental issues. Land and forest fires in peatland areas have caused various conflicts with extensive negative impacts—technically, ecologically, economically, socially, and culturally [5]—such as (1) peatland fires caused by misuse, carelessness, and neglect, and intentionally; (2) dry peats formed by creating canals and planting non-peat-friendly plants; (3) damage to peatland; and (4) decreased productivity of peatlands. Such conditions lead to negative economic impacts, such as loss of livelihoods and decreased incomes.

Forest loss in Indonesia has continued to increase since 2002, reaching the highest loss of more than 900,000 ha in 2016 due to the forest fires in 2015 [6]. Much of the forest loss in the period was within areas classified as secondary forest and other land cover (for example, mixed dry land agriculture, estate crop, plantation forest, shrub, and others) [7,8]. Forest loss decreased from then until 2022. However, forest loss in 2022 still reached over 100,000 ha [9].

Vulnerability is determined by physical, social, economic, and environmental factors or processes in a community and by the impact of hazards [10]. Vulnerability is a condition influenced by physical, social, economic, and environmental processes that can increase the risk of the impact of a hazard [11]. In general terms, vulnerability is a condition where the system cannot adjust to the impact of a change [12]. The nature of vulnerability differs

temporally and spatially [13,14]. Vulnerabilities can be divided based on impact, such as those related to economic, social, ecological, livelihood, and environmental aspects. According to [11], vulnerability in a social context is a function of exposure, adaptive capability, and sensitivity. Community vulnerability is a condition in which a community cannot adapt to ecosystem changes caused by a particular threat [15]. From an economic perspective, vulnerability includes population and institutional vulnerability depending on the existence of institutions in the area or the village. Vulnerability factors include the following [16]: (1) physical vulnerability: basic infrastructure, construction, buildings; (2) economic vulnerability: poverty, income, nutrition; (3) social vulnerability: education, health, politics, legal, institutional; and (4) environmental vulnerability: soil, water, plants, forests, oceans.

In addition, vulnerability can also affect the welfare of a community, whereby the greatest impact can be seen from shifting or reducing livelihoods [17,18]. Improving people's livelihoods on peatland through developing business opportunities is important and inherent in the understanding of the vulnerability of the people who do business in and/or around the peat ecosystem who are affected by changes to the ecosystem [19].

This study aims to describe, measure, and analyze the level of vulnerability of farm households due to land and forest fires in peatland areas and observe the changes in ecosystems in those areas in three peatland hydrological units (PHUs) in Ogan Komering Ilir (OKI) District, South Sumatra Province, Indonesia. It is expected that outputs from this research will improve understanding of the levels of social, economic, livelihood, ecological, and climate vulnerability. The study also assists with mapping community conditions based on the distribution of levels of vulnerability and provides indicators for interventions to address vulnerability in the affected areas.

## 2. Materials and Methods

### 2.1. Study Sites

OKI District is one of four peat restoration priority districts in South Sumatra. The district includes five PHUs with an estimated area of 1,108,483.41 ha. The names of the five PHUs as the study areas are presented in Figure 1.

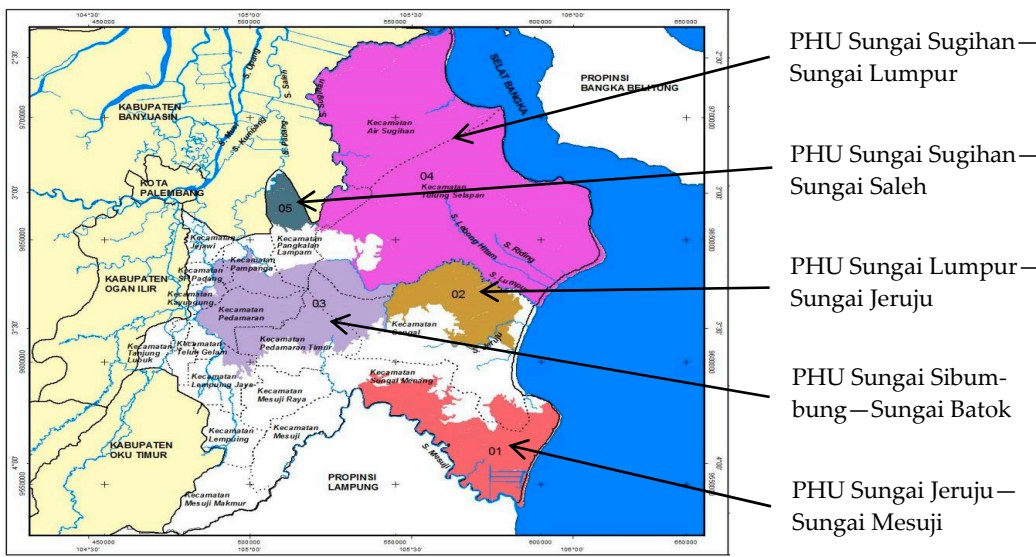

**Figure 1.** Locations and areas of PHUs in OKI District.

Given the large size of the study area and the large number of affected households, this study was carried out using a household sample survey method and three approaches: (1) PHU approach; (2) administrative area approach; and (3) activity space approach.

*2.2. Sampling and Data Collection*

Sampling was carried out using a cluster sampling method with three sampling stages: (1) determining the PHU; (2) selecting sub-districts and sample villages; and (3) selecting household samples. The description of the sampling follows.

1. Of the five PHUs in OKI District, three were selected based on the variety of natural resources (including peatland) and the diversity of people's livelihoods: (1) PHU Sungai Sugihan–Sungai Lumpur; (2) PHU Sungai Sibumbung–Sungai Batok; (3) PHU Sungai Saleh–Sungai Sugihan.

2. In each PHU, sub-district and village clusters were determined based on the main livelihood of the population, for example, sub-district and village clusters with the main livelihood of the population being food crop farming (rice, other crops, horticulture), plantation crop clusters (rubber, oil palm, etc.), forest plant clusters and non-timber forest product (NTFP) collection, livestock clusters (swamp buffalo, cows/goats, chickens/ducks), fishery clusters (aquaculture, capture), home industry clusters/small processing industries, service clusters, and others.

3. From each sub-district and village cluster, two sample villages were selected representing the characteristics of the cluster.

4. Stratified random sampling was conducted in each village based on the area of cultivated land (for the livelihoods of crop and estate farming), number of livestock, number of business units (fisheries), production amount (timber collection and NTFPs), ownership of assets (manufacturing industry), etc. The sample characteristics within each livelihood type are quite homogeneous, such that the number of sample households drawn was adjusted to their respective populations.

For households whose main livelihoods were outside the village area, for example, looking for wood and NTFPs, the sampling was carried out in their home area not at their work location. In this case, the spatial mobility of the population was considered in relation to the impact of livelihoods on the peat ecosystem.

Upon random selection, household respondents were then interviewed, which was followed by an in-depth interview as necessary. In addition, field observations were also conducted to confirm the data collected during the interview. Furthermore, focus group discussions (FGDs) were implemented to clarify and triangulate some important and specific findings.

*2.3. Data Processing and Measurement of Vulnerability*

Data obtained through this study were processed using descriptive analysis, namely, calculating the average sample value (mean, median or mode, and standard deviation). The level of household vulnerability was measured with scores for indicators obtained from the survey. The vulnerability level is presented in tables and graphs for easy interpretation and comparison.

2.3.1. Social Vulnerability

Social vulnerability is a condition in which a household is in a state of vulnerability as shown by several household social indicators [20]. In this study, social vulnerability was measured using scores for five indicators: (1) number of household members [21]; (2) number of children under five (including infants) and elderly in the household [22]; (3) residential status, that is, whether a local resident or a migrant; (4) length of stay; and (5) poverty status [23].

In our study, social vulnerability was divided into three groups. Household vulnerability was categorized as high if there were three or more members aged under 5 and elderly members of one or more; moderate if there were one to two members aged under 5 and elderly members of one or more; and low if there were no children under 5 nor any elderly members. Migrant households were categorized as high vulnerability, while local

residents were rated as low vulnerability since the latter were easily supported by families who lived nearby when facing a vulnerable situation.

In terms of length of residence, household vulnerability was categorized as high if resident for 20 years or less; moderate if resident for up to 40 years; and low if resident for more than 40 years. Likewise, household vulnerability was categorized as high if the household fell into the "poor" group and low if not.

### 2.3.2. Economic Vulnerability

Economic vulnerability is a condition in which a household is in a state of vulnerability as measured by several indicators [24]. In our study, we used scores for five indicators: (1) household income; (2) household per capita income; (3) household expenditure; (4) business land ownership; and (5) condition of the housing. Household income was estimated using both financial income (e.g., from selling the products) and the products that were self-consumed (subsistence). Based on household income, the level of household economic vulnerability was divided into three classes: (1) low vulnerability if household income was greater than IDR 3,500,000 per month; (2) moderate vulnerability if it was between IDR 1,750,000 and IDR 3,500,000 per month, (3) high vulnerability if it was IDR 1,750,000 per month or less.

Based on the per capita income, the household economic vulnerability was divided into 3 classes, namely: (1) low vulnerability if per capita income was greater than IDR 750,000 (≈USD 48) per month; (2) moderate vulnerability if it was between IDR 370,000 (≈USD 24) and IDR 750,000 per month; and (3) high vulnerability if it was IDR 370,000 per month or less.

Household expenditure per month was also divided into three classes: (1) low vulnerability if expenditure was greater than IDR 1,500,000 (≈USD 96) per month; (2) medium vulnerability if it was between IDR 1,000,000 (≈USD 64) and IDR 1,500,000 per month; and (3) high vulnerability if it was IDR 1,000,000 per month or less.

Based on business land ownership, household economic vulnerability was also divided into three classes: (1) low vulnerability if business land ownership was larger than 1.0 ha; (2) moderate vulnerability if it was between 0.5 and 1.0 ha; and (3) high vulnerability if it was 0.5 ha or less [23,24].

The condition of housing was also divided into three classes: (1) low vulnerability if permanent housing; (2) moderate vulnerability if semi-permanent housing; and (3) high vulnerability if emergency housing.

### 2.3.3. Livelihood Vulnerability

A household's livelihood vulnerability [25] was measured using scores for four indicators of livelihoods applied to the household head and/or household members: (1) the main type of livelihood of the household head; (2) the length of time (in months) the household head worked in a year; (3) the education level of the household head; and (4) the number of household members who were working.

Respondents were divided into three groups: (1) farmers, fishers, and laborers as a group with a high level of vulnerability due to the seasonal nature of their livelihoods; (2) planters, traders, and entrepreneurs as a group with a moderate level of vulnerability; and (3) employers/employees as a group with a low level of vulnerability.

The working period of the head of the household in a year (in months) was also grouped in three classes: (1) working up to 4 months was categorized as high vulnerability; (2) working 5 to 8 months was categorized as moderate vulnerability; and (3) working 9 to 12 months as low vulnerability.

The education level of the head of the household was divided into three groups: (1) primary school education was categorized as having high vulnerability; (2) secondary school education was moderate vulnerability; and (3) undergraduate education was low vulnerability.

The number of working household members (other than the head of the household) was also grouped into three: (1) if there were no working household members, household

vulnerability was categorized as high; (2) if there was one working household member, vulnerability was moderate; and (3) if there were two or more working household members, it was categorized as low vulnerability.

### 2.3.4. Ecological Vulnerability

Ecological vulnerability is a condition in which a household is in a state of vulnerability, as shown by several indicators registering negative changes (damage or deterioration) in ecosystem components, including land, water, plantations, and the availability of NTFPs [26,27]. The damage or deterioration of ecosystem components was measured based on the opinion of the respondents, using the following criteria: (1) if there was no change or slight damage to land, water, or crops, then the ecological vulnerability was categorized as low; (2) if there was moderate damage, then it was categorized as moderate; and (3) if there was severe damage, then it was categorized as high.

In terms of changes in resource availability, the level of ecological vulnerability was measured using the following criteria: (1) if the availability of resources was constant, then ecological vulnerability was considered to be low; (2) if resource availability was reduced, it was moderate; and (3) if resource availability was very highly reduced, then it was considered to be highly vulnerable.

### 2.3.5. Climate Change Vulnerability

Climate change vulnerability is measured by the impact of climate change on people's livelihoods [28,29]. In our study, we measured two types of climate change impacts (drought and floods) and four types of community livelihoods (agriculture, plantation, animal husbandry, and forestry) resulting in eight climate change indicators. We measured based on community respondents' observations of changes that had occurred: (1) if there was no change or a slight change/impact, then it was categorized as low; (2) if there was a moderate level of change, then it was categorized as moderate; and (3) if there were severe changes, it was categorized as high vulnerability.

## 3. Results

### 3.1. Social Vulnerability

Considering the "number of household members" and "number of children under 5 and the elderly" indicators, results showed that most of the sample households in the three PHUs were at a moderate level of social vulnerability.

Based on the "poor" indicator, the majority of sample households in PHU Sungai Sebumbung–Sungai Batok and PHU Sungai Sugihan–Sungai Lumpur were at a low level of social vulnerability, while in PHU Sungai Saleh–Sungai Sugihan, the distribution of low and high levels of social vulnerability was the same (Table 1).

**Table 1.** Results of social vulnerability measurement.

| No. | Indicator | Level of Social Vulnerability (%) | | | Average Score |
|---|---|---|---|---|---|
| | | Low | Medium | High | |
| | PHU S. Sebumbung–S. Batok | | | | |
| 1 | Number of household members | 10.0 | 48.0 | 42.0 | 2.32 |
| 2 | Number of toddlers and elderly | 49.0 | 48.0 | 3.0 | 1.54 |
| 3 | Resident status | 87.0 | 0 | 13.0 | 1.26 |
| 4 | Length of stay | 47.0 | 47.0 | 6.0 | 1.59 |
| 5 | Poverty status | 80.0 | 0 | 20.0 | 1.40 |
| | Total score (interval 5–15) | | | | 8.11 |

**Table 1.** *Cont.*

| No. | Indicator | Level of Social Vulnerability (%) | | | Average Score |
|---|---|---|---|---|---|
| | | Low | Medium | High | |
| | PHU S. Saleh–S. Sugihan | | | | |
| 1 | Number of household members | 5.0 | 64.0 | 31.0 | 2.26 |
| 2 | Number of toddlers and elderly | 25.0 | 63.0 | 12.0 | 1.87 |
| 3 | Resident status | 49.0 | 46.0 | 5.0 | 1.56 |
| 4 | Length of stay | 95.0 | 0.0 | 5.0 | 1.10 |
| 5 | Poverty status | 49.0 | 0.0 | 51.0 | 2.02 |
| | Total score (interval 5–15) | | | | 8.81 |
| | PHU S. Sugihan–S. Lumpur | | | | |
| 1 | Number of household members | 9.0 | 81.0 | 10.0 | 2.01 |
| 2 | Number of toddlers and elderly | 47.0 | 47.0 | 6.0 | 1.59 |
| 3 | Resident status | 0.0 | 84.0 | 16.0 | 2.16 |
| 4 | Length of stay | 35.0 | 0.0 | 65.0 | 2.30 |
| 5 | Poverty status | 80.0 | 0.0 | 20.0 | 1.40 |
| | Total score (interval 5–15) | | | | 9.46 |
| | Average score for all PHUs | | | | 8.79 |

When compared among the three PHUs, the highest social vulnerability score was observed for PHU Sungai Sugihan–Sungai Lumpur, while the lowest vulnerability was observed for PHU Sungai Sebumbung–Sungai Batok. Differences in social vulnerability among the three PHUs were observed mainly for the indicators "length of stay" and "residential status". In terms of the indicators "number of household members" and the "number of children under five and the elderly", there were no significant differences among the three PHUs (Figure 2).

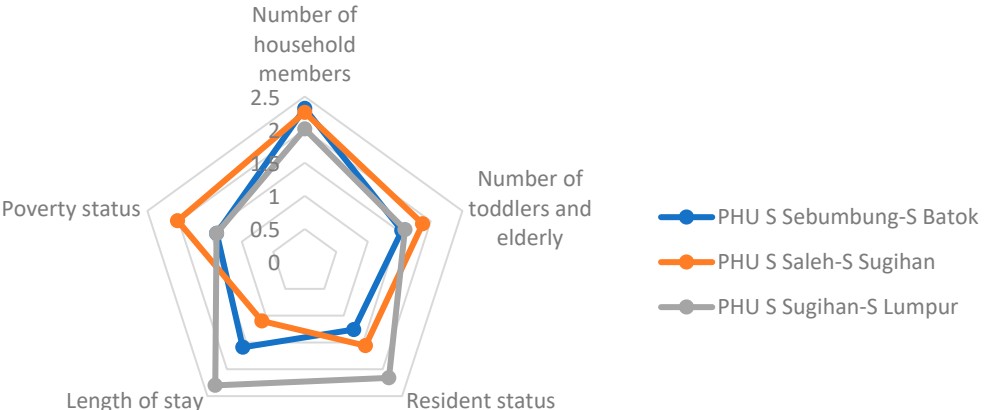

**Figure 2.** Social vulnerability score based on indicators.

*3.2. Economic Vulnerability*

Considering "household income", "per capita income", and "household expenditure" indicators, results show that economic vulnerability is relatively even in PHU Sungai Sebumbung–Sungai Batok. In PHU Sungai Saleh–Sungai Sugihan, the percentage of high vulnerability is greater than that of medium and low vulnerability. In PHU Sungai Sugihan–Sungai Lumpur, based on household income indicators, most households are at a high level of vulnerability (Table 2).

**Table 2.** Results of economic vulnerability measurement.

| No. | Indicator | Level of Economic Vulnerability (%) | | | Average Score |
|-----|-----------|------|--------|------|---------------|
| | | Low | Medium | High | |
| | PHU S. Sebumbung–S. Batok | | | | |
| 1 | Household income | 330 | 34.0 | 33.3 | 2.00 |
| 2 | Income per capita | 32.0 | 36.0 | 32.0 | 2.00 |
| 3 | Household expenses | 27.0 | 41.0 | 32.0 | 2.05 |
| 4 | Land ownership | 12.0 | 53.0 | 35.0 | 2.23 |
| 5 | Home conditions | 53.0 | 44.0 | 3.0 | 1.50 |
| | Total score (interval 5–15) | | | | 9.78 |
| | PHU S. Saleh–S. Sugihan | | | | |
| 1 | Household income | 22.0 | 33.0 | 45.0 | 2.23 |
| 2 | Income per capita | 22.0 | 33.0 | 45.0 | 2.23 |
| 3 | Household expenses | 24.0 | 31.0 | 45.0 | 2.21 |
| 4 | Land ownership | 9.0 | 28.0 | 63.0 | 2.54 |
| 5 | Home conditions | 24.0 | 73.0 | 3.0 | 1.79 |
| | Total score (interval 5–15) | | | | 11.00 |
| | PHU S. Sugihan–S. Lumpur | | | | |
| 1 | Household income | 29.0 | 0.0 | 71.0 | 2.42 |
| 2 | Income per capita | 32.0 | 36.0 | 32.0 | 2.00 |
| 3 | Household expenses | 22.0 | 45.0 | 33.0 | 2.11 |
| 4 | Land ownership | 32.0 | 35.0 | 33.0 | 2.01 |
| 5 | Home conditions | 62.0 | 27.0 | 11.0 | 1.49 |
| | Total score (interval 5–15) | | | | 10.03 |
| | Average score for all PHUs | | | | 10.27 |

When compared among the three PHUs, the highest economic vulnerability score was observed for PHU Sungai Saleh–Sungai Sugihan. The difference in economic vulnerability scores between PHU Sungai Sugihan–Sungai Lumpur and PHU Sungai Sebumbung–Sungai Batok was not significant. The difference in economic vulnerability scores between PHU Sungai Saleh–Sungai Sugihan and the other two PHUs was mainly found in the land ownership and home conditions indicators (Figure 3).

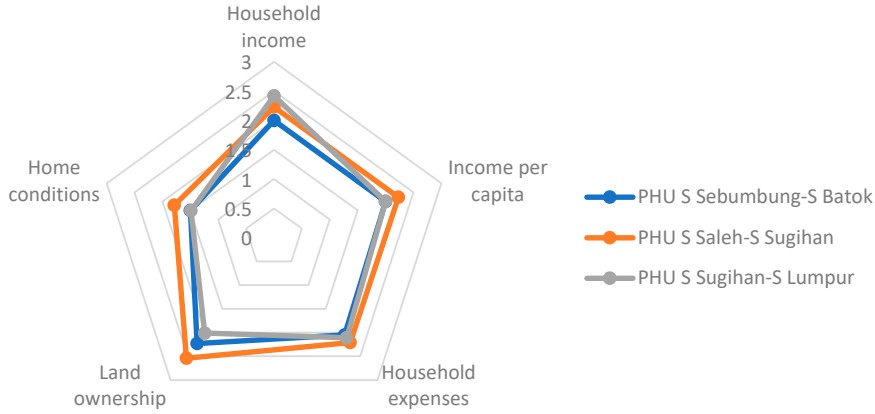

**Figure 3.** Economic vulnerability score based on indicators.

*3.3. Livelihood Vulnerability*

Considering the "household head's main occupation" indicator, results show that livelihood vulnerability level was high in the three PHUs, especially in PHU Sungai Saleh–Sungai Sugihan.

The level of livelihood vulnerability in the three PHUs is also high based on the "household head's education" indicator, especially in PHU Sungai Saleh–Sungai Sugihan.

However, the level of livelihood vulnerability in PHU Sungai Saleh–Sungai Sugihan based on "the amount of working months" and "the number of working household members" indicators is the lowest among the three PHUs (Table 3).

**Table 3.** Results of livelihood vulnerability measurement.

| No. | Indicator | Level of Livelihood Vulnerability (%) | | | Average Score |
|---|---|---|---|---|---|
| | | Low | Medium | High | |
| | PHU S. Sebumbung–S. Batok | | | | |
| 1 | Household head's main occupation | 2.0 | 44.0 | 54.0 | 2.52 |
| 2 | Number of working months in a year | 8.0 | 55.0 | 37.0 | 2.29 |
| 3 | Household head's education | 2.0 | 40.0 | 58.0 | 2.56 |
| 4 | Number of working household members | 28.0 | 39.0 | 33.0 | 2.05 |
| | Total score (interval 4–12) | | | | 9.42 |
| | PHU S. Saleh–S. Sugihan | | | | |
| 1 | Household head's main occupation | 3.0 | 1.0 | 96.0 | 2.93 |
| 2 | Number of working months in a year | 100.0 | 0.0 | 0.0 | 1.00 |
| 3 | Household head's education | 4.0 | 23.0 | 73.0 | 2.69 |
| 4 | Number of working household members | 72.0 | 28.0 | 0.0 | 1.28 |
| | Total score (interval 4–12) | | | | 7.90 |
| | PHU S. Sugihan–S. Lumpur | | | | |
| 1 | Household head's main occupation | 0.0 | 45.0 | 55.0 | 2.55 |
| 2 | Number of working months in a year | 51.0 | 42.0 | 7.0 | 1.56 |
| 3 | Household head's education | 3.0 | 38.0 | 59.0 | 2.56 |
| 4 | Number of working household members | 38.0 | 62.0 | 0.0 | 1.62 |
| | Total score (interval 4–12) | | | | 7.29 |
| | Average score for all PHUs | | | | 8.20 |

When compared among the three PHUs, the highest livelihood vulnerability score was observed for PHU Sungai Sebumbung–Sungai Batok. Between PHU Sungai Saleh–Sungai Sugihan and PHU Sungai Sugihan–Sungai Lumpur, the level of livelihood vulnerability was only slightly different. PHU Sungai Saleh–Sungai Sugihan had the lowest livelihood vulnerability score among the three PHUs. Differences in livelihood vulnerability between the three PHUs are mainly found in "the number of working household members" and "the number of working months" indicators (Figure 4).

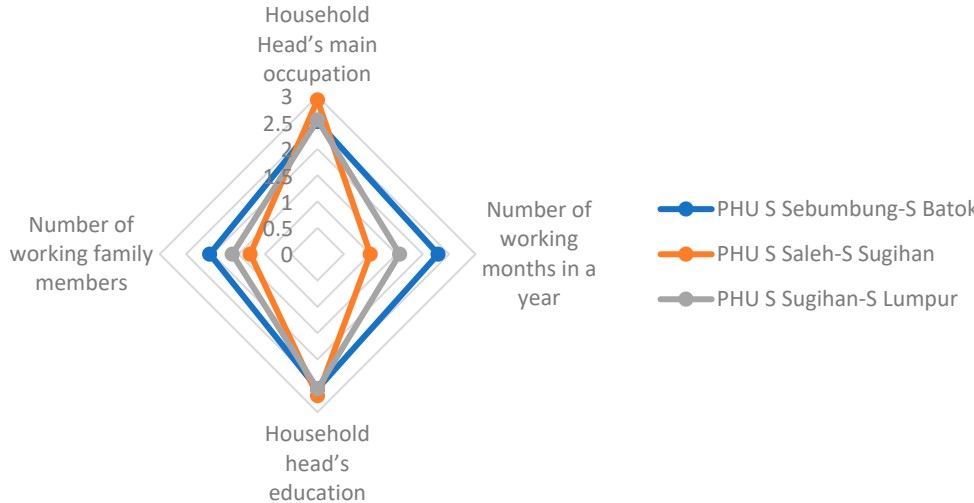

**Figure 4.** Livelihood vulnerability score based on indicators.

*3.4. Ecological Vulnerability*

Considering the "damage to soil", the "damage to water", and the "damage to cultivation" indicators, results show that the ecological vulnerability in the three PHUs is relatively low. The indicator of ecological vulnerability is considered moderate based on the availability of NTFPs, especially in PHU Sungai Saleh–Sungai Sugihan and PHU Sungai Sugihan–Sungai Lumpur (Table 4).

**Table 4.** Results of ecological vulnerability measurement.

| No. | Indicator | Level of Ecological Vulnerability (%) | | | Average Score |
|---|---|---|---|---|---|
| | | Low | Medium | High | |
| | PHU S. Sebumbung–S. Batok | | | | |
| 1 | Damage to soil | 85.0 | 6.0 | 9.0 | 1.24 |
| 2 | Damage to water | 90.0 | 6.0 | 4.0 | 1.14 |
| 3 | Damage to cultivation | 98.0 | 2.0 | 0.0 | 1.02 |
| 4 | Availability of non-timber forest products | 59.0 | 41.0 | 0.0 | 1.41 |
| | Total score (interval 4–12) | | | | 4.81 |
| | PHU S. Saleh–S. Sugihan | | | | |
| 1 | Damage to soil | 65.0 | 24.0 | 11.0 | 1.46 |
| 2 | Damage to water | 69.0 | 19.0 | 12.0 | 1.43 |
| 3 | Damage to cultivation | 59.0 | 17.0 | 24.0 | 1.65 |
| 4 | Availability of non-timber forest products | 11.0 | 60.0 | 29.0 | 2.18 |
| | Total score (interval 4–12) | | | | 6.72 |
| | PHU S. Sugihan–S. Lumpur | | | | |
| 1 | Damage to soil | 93.0 | 4.0 | 3.0 | 1.10 |
| 2 | Damage to water | 92.0 | 7.0 | 1.0 | 1.09 |
| 3 | Damage to cultivation | 87.0 | 12.0 | 1.0 | 1.14 |
| 4 | Availability of non-timber forest products | 28.0 | 67.0 | 5.0 | 1.77 |
| | Total score (interval 4–12) | | | | 5.10 |
| | Average score for all PHUs | | | | 5.54 |

When compared among the three PHUs, the highest ecological vulnerability score was observed for PHU Sungai Saleh–Sungai Sugihan and the lowest ecological vulnerability was observed for PHU Sungai Sebumbung–Sungai Batok. PHU Sungai Saleh–Sungai Sugihan had the highest ecological vulnerability, based on the all four indicators. The four indicators of ecological vulnerability are consistent in ranking the ecological vulnerability of the three PHUs (Figure 5).

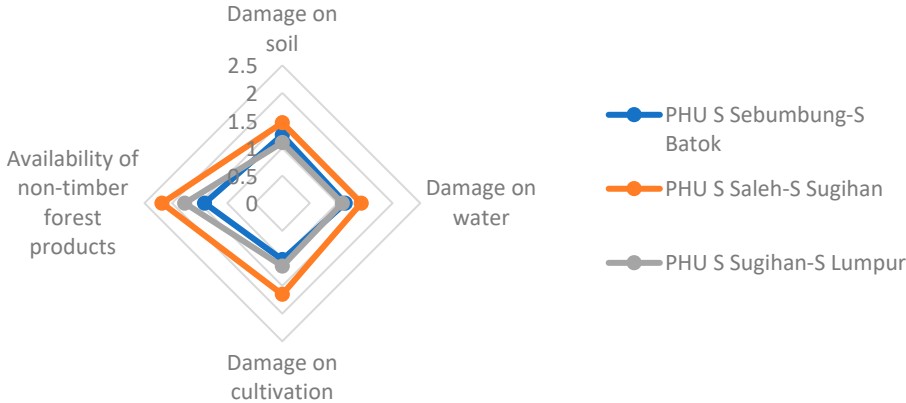

**Figure 5.** Ecological vulnerability score based on indicators.

### 3.5. Climate Change Vulnerability

The results show that vulnerability due to climate change in the three PHUs is low, based on all the indicators, except the "flooding in the agricultural sector" and the "drought in the agricultural sector" indicators. Flooding has an impact on the vulnerability of the agricultural sector in PHU Sungai Sebumbung–Sungai Batok, while drought has an impact on the vulnerability of the agricultural sector in PHU Sungai Saleh–Sungai Sugihan (Table 5).

When compared among the three PHUs, the highest climate change vulnerability was observed for PHU Sungai Sebumbung–Sungai Batok, followed by PHU Sungai Saleh–Sungai Sugihan. PHU Sungai Sugihan–Sungai Lumpur has the lowest climate change vulnerability score among the three PHUs. Differences in climate change vulnerability between the three PHUs were mainly found in the "drought for agriculture" and "flood for agriculture" indicators. The influence of drought indicators on plantations only occurs in PHU Sungai Saleh–Sungai Sugihan (Figure 6).

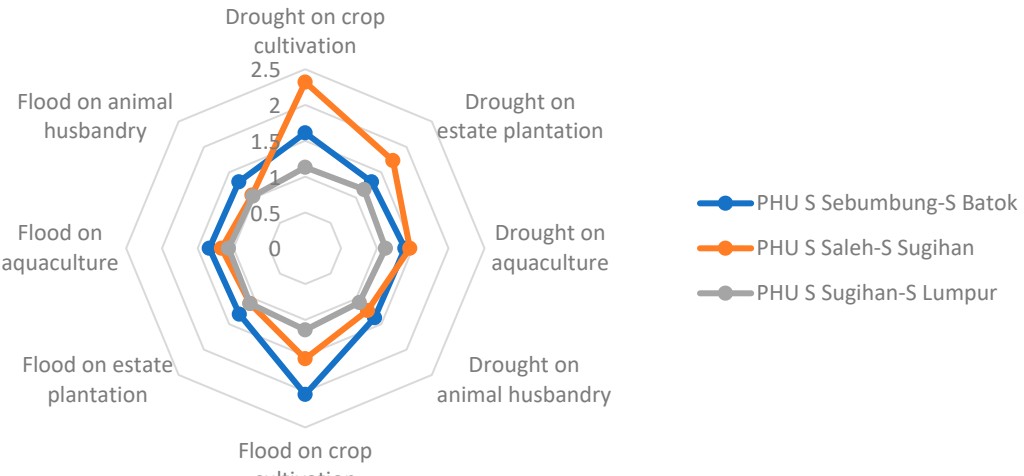

**Figure 6.** Climate change vulnerability score based on indicators.

**Table 5.** Results of climate change vulnerability indicators.

| No. | Indicator | Level of Climate Change Vulnerability (%) | | | Average Score |
|---|---|---|---|---|---|
| | | Low | Medium | High | |
| | PHU S. Sebumbung–S. Batok | | | | |
| 1 | Drought in crop cultivation | 57.0 | 25.0 | 18.0 | 1.61 |
| 2 | Drought in estate plantation | 79.0 | 11.0 | 10.0 | 1.31 |
| 3 | Drought in aquaculture | 74.0 | 13.0 | 13.0 | 1.39 |
| 4 | Drought in animal husbandry | 77.0 | 9.0 | 14.0 | 1.37 |
| 5 | Flood in crop cultivation | 30.0 | 36.0 | 34.0 | 2.04 |
| 6 | Flood in estate plantation | 81.0 | 8.0 | 11.0 | 1.30 |
| 7 | Flood in aquaculture | 78.0 | 10.0 | 12.0 | 1.34 |
| 8 | Flood in animal husbandry | 80.0 | 9.0 | 11.0 | 1.31 |
| | Total score (interval 8–24) | | | | 11.67 |
| | PHU S. Saleh–S. Sugihan | | | | |
| 1 | Drought in crop cultivation | 24.0 | 20.0 | 56.0 | 2.32 |
| 2 | Drought in estate plantation | 50.0 | 27.0 | 23.0 | 1.73 |
| 3 | Drought in aquaculture | 68.0 | 18.0 | 14.0 | 1.46 |
| 4 | Drought in animal husbandry | 83.0 | 11.0 | 6.0 | 1.23 |
| 5 | Flood in crop cultivation | 64.0 | 18.0 | 18.0 | 1.54 |
| 6 | Flood in estate plantation | 92.0 | 7.0 | 1.0 | 1.09 |
| 7 | Flood in aquaculture | 85.0 | 13.0 | 2.0 | 1.17 |
| 8 | Flood in animal husbandry | 95.0 | 5.0 | 0.0 | 1.05 |
| | Total score (interval 8–24) | | | | 11.59 |
| | PHU S. Sugihan–S. Lumpur | | | | |
| 1 | Drought in crop cultivation | 89.0 | 9.0 | 2.0 | 1.13 |
| 2 | Drought in estate plantation | 86.0 | 12.0 | 2.0 | 1.16 |
| 3 | Drought in aquaculture | 89.0 | 10.0 | 1.0 | 1.12 |
| 4 | Drought in animal husbandry | 94.0 | 5.0 | 1.0 | 1.07 |
| 5 | Flood in crop cultivation | 87.0 | 12.0 | 1.0 | 1.14 |
| 6 | Flood in estate plantation | 91.0 | 9.0 | 0.0 | 1.09 |
| 7 | Flood in aquaculture | 93.0 | 7.0 | 0.0 | 1.07 |
| 8 | Flood in animal husbandry | 97.0 | 3.0 | 0.0 | 1.03 |
| | Total score (interval 8–24) | | | | 8.81 |
| | Average score all PHUs | | | | 10.69 |

## 4. Discussion

In this study, we measured and analyzed vulnerability in five categories: social, economic, livelihood, ecological, and climate change. First, for social vulnerability, this study showed that among the three PHUs, the highest vulnerability score was observed for PHU Sungai Sugihan–Sungai Lumpur, while the lowest was observed for PHU Sungai Sebumbung–Sungai Batok. The data also show, based on the residency indicator, that the PHU Sungai Sugihan area had the highest social vulnerability level. The results showed that the population domiciled in this area is dominated by indigenous people, with some

working as day laborers in large, company-controlled, land concession areas to fulfill their daily needs [30].

Previous research in Banjar Baru, Kalimantan found that social vulnerability was high in a community in an area prone to fire. Overcrowding caused by the large number of household members in an area also leads to greater social vulnerability. Increased public awareness, including an understanding of the causes, and handling of, disasters can help reduce social vulnerability [31]. Overall, in an effort to reduce social vulnerability in a fire-prone area, physical restoration activities, such as canal blocking and canal back-filling, will be useful [2,30].

Second, for economic vulnerability, we found that the highest economic vulnerability score occurred in PHU Sungai Saleh–Sungai Sugihan. Meanwhile, between PHU Sungai Sugihan–Sungai Lumpur and PHU Sungai Sebumbung–Sungai Batok, the difference in the level of economic vulnerability was not great. This finding aligns with the "household income" indicator, in which the residents of PHU Sungai Sugihan–Sungai Lumpur area fall into the highest economic vulnerability category. One of the reasons for this could be because some residents undertake day laboring in concession areas as their main job to meet their daily needs [30].

In areas outside large company concessions, peat restoration activities will be easier to implement because the Government can directly carry out restoration activities in those areas [32,33], unlike the company-controlled concession areas wherein the responsibility for any restoration falls to the company. Nevertheless, restoration indirectly has an impact on the income of farming households in a concession area because an increase in farming household income on peatland has been shown to be strongly influenced by restoration activities [34–36]. However, peatland restoration activities cannot take place without collaboration and coordination among several related parties [37].

In addition to farmers' household income, the area of land owned by farmers can also determine the degree of economic vulnerability [38], as experienced by farmers' groups in the PHU Sungai Saleh–Sungai Sugihan area where economic vulnerability measured based on the area of land owned was in the highest category. The larger the land area, the higher the net income that will be received by farmers [39,40].

Based on the indicator "household head employment", PHU Sungai Saleh–Sungai Sugihan had high economic vulnerability wherein the head of a household worked as a farmer, fisher, or laborer, occupations that are highly dependent on natural conditions.

In line with the results of this research, natural capital-based livelihood strategies that use existing natural resources combined with agricultural cultivation are the main strategies chosen by the community to sustain their livelihoods [41]. Nevertheless, the use of peatland for agricultural activities has its own challenges, including fires, soil acidity, low fertility, and limited choice of suitable species [42]. Some of these challenges increase the risk of the income of the head of the household being uncertain. To reduce income uncertainties, it is important to have specially designed farming systems and patterns that can provide direct and multiple benefits to the local community.

Agrosilvofishery is an agricultural system that can be applied on peatland to reduce income uncertainties. The system combines different activities, including agriculture (such as agroforestry and small-scale farming), aquaculture (fish farming), and forestry (sustainable timber extraction), to create a multi-functional and sustainable system [43]. Agrosilvofishery is not just an agricultural system; it can also diversify and enhance the various livelihood practices on peatland and has the potential to reduce income uncertainty or risk and improve household welfare and food security through diversifying livelihoods [44,45].

Some countries with peatland areas have implemented integrated approaches such as agrosilvofishery systems more extensively than others. For example, in Bangladesh, agrosilvofishery is promoted to enhance agricultural productivity and rural livelihoods [46–48]. In certain regions of China, agrosilvofishery practices are implemented to improve sustainable land use and enhance agricultural productivity. Examples include integrating

aquaculture with wetland agriculture or incorporating fish production in rice fields [49–51]. Agrosilvofishery practices are also promoted in Costa Rica as part of sustainable agricultural systems in which combining agricultural activities with reforestation efforts and fish production is encouraged [52].

Ecological vulnerabilities can be divided into those caused by natural or human factors [53]. However, most of the research on ecological vulnerability in peatland areas has considered only natural factors [54]. Our study considers ecological vulnerability caused by both human and natural factors.

Ecological vulnerability assessment is an effective tool to alleviate contradictions [55]. The different assessment in our study compared with that of others shows the role of human society in changing inherent natural ecological vulnerability [55]. For example, land destruction can occur due to land disturbance in peatland [53]. This can occur naturally due to the El Niño–Indian Ocean Dipole phenomenon or because of humans who deliberately set fires to clear land. One of the impacts of fire is that it can lead to higher acidity levels. This will certainly be very detrimental to farmers because they have to spend more to prepare the land for cultivation [56].

Evaluating ecological vulnerability is significant for protecting and promoting ecosystem stability. However, attention to the dimensions of vulnerability and socio-ecological risk is lacking, indicating a large knowledge gap, especially when considering that environmental degradation is considered one of the main causes of natural disaster risk worldwide [57]. As an effort to reduce ecological vulnerability, one of the adaptable frameworks that can be applied is to overcome the driving factors of unwanted ecological changes caused by humans. In addition, to implement effective, long-term, and sustainable behavioral adaptation, there needs to be a greater emphasis on strategies that are capable of improving human values, skills, and behaviors. In other words, a participatory approach to environmental management could be part of the solution to reduce the percentage of ecological vulnerability [58].

In previous studies, climate change vulnerabilities were measured using indicators such as drought, temperature increase, pests, and land degradation. However, in this study, climate change vulnerability that occurs in four agribusiness sub-sectors—crop cultivation, aquaculture, estate plantation, and animal husbandry—have a low climate change vulnerability category.

Some of the causes of climate change vulnerability, especially in peatlands, include (1) farmers' lack of knowledge and information related to the phenomenon of climate change; (2) weakness of farmers' memory in monitoring climate change; [59] and (3) the fact that climate change does not occur instantly but continuously. If left unaddressed, droughts and floods will have a long-term negative impact, including environmental damage, decreased productivity of agricultural, plantation, fishery and livestock products, and crop failure. This will certainly increase the economic vulnerability of farming households because the damage will reduce farmers' household income, especially that of small-scale and subsistence farmers [60].

There is a need for integration and implementation of climate change adaptation policies in local government operations to reduce the vulnerability of smallholders and increase their ability to absorb, adapt, and transform in the face of climate change [61]. In addition, other forms of adaptation strategies that can be applied by farmers would be using superior seed, adjusting planting patterns and times, and carrying out water management and fish farming techniques that are suitable all-year round [62].

## 5. Conclusions and Implications

The results of this study led to the following conclusions:

1.  Conflicts that often occur in the management of livelihoods on peatland are more related to the use of natural resources and ecological limitations in meeting human needs since the livelihoods of local people were still dependent on the availability of natural resources in the peatland areas and their surrounds.

2.　Vulnerability scores vary by the type of vulnerability and PHU. PHU Sungai Sebumbung–Sungai Batok had the highest score for livelihood and climate change vulnerability, but the lowest for social, economic, and ecological vulnerability. PHU Sungai Saleh–Sungai Sugihan had the highest score for economic and ecological vulnerability, but the lowest for livelihood vulnerability. PHU Sungai Sugihan–Sungai Lumpur had the highest score for social vulnerability, but the lowest for climate change vulnerability.

3.　The indicators "number of household members" and "number of children under 5 and the elderly" make relatively equal contributions to the social vulnerability score in the three PHUs. All economic indicators except "business land ownership" make relatively equal contributions to the economic vulnerability score in the three PHUs. The indicator "length of time a household works in a year" is an important indicator in determining variations in livelihood vulnerability among the three PHUs. Sungai Saleh–Sungai Sugihan is the PHU with the highest ecological vulnerability score for all vulnerability indicators. The agricultural sector has the highest vulnerability due to the impact of climate change, such as droughts and floods.

The following implications are proposed for mitigating vulnerability before it becomes severe and difficult to tackle:

1.　Development of various alternatives of resource-based local livelihoods, such as processing buffalo milk into various products, processing local fish into smoked and salted fish, processing *purun* (*Eleocharis dulcis*) (in partnership with companies) to improve living standards, and reducing the need for annual burning.

2.　Community involvement in resource management and fire prevention is seen as an effective way to prevent forest and peatland fires. This can be implemented through provision of socio-economic incentives to communities for sustainable management of peatland, creating and strengthening local institutions and maintaining regulations for fire management.

3.　Provision of social back-up in times of crisis due to land and forest fire.

4.　Development of formal institutions to support the processing of local resources into various products, such as buffalo milk products, smoked and salted fish, and *purun*-based products.

5.　Development of markets to ensure that economic activities can result in an increase in household income and welfare.

6.　Inclusion of alternative strategies that households do or should do in coping with the difficulties caused by land and forest fire based on their past experience.

**Author Contributions:** Conceptualization, M.Y.; Methodology, M.Y., D.A. and R.; Software, D.D.; Validation, D.A. and D.D.; Formal analysis, M.Y. and R.; Investigation, M.Y., D.A. and R.; Resources, D.D.; Writing—original draft, M.Y.; Writing—review & editing, D.A., R. and D.D.; Visualization, D.D.; Supervision, M.Y. All authors have read and agreed to the published version of the manuscript.

**Funding:** The APC was funded by the CIFOR-ICRAF.

**Data Availability Statement:** The data presented in this study are available on request from the corresponding author. The data are not publicly available due to privacy restrictions.

**Conflicts of Interest:** The authors declare no conflict of interest.

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
