# Peer review of "Farm Household Vulnerability Due to Land and Forest Fire in Peatland Areas in South Sumatra"

_land, doi:10.3390/land13050642_

Round 1

Reviewer 1 Report

Comments and Suggestions for Authors

Dear authors, 

Congratulations to your manuscript. I found the topic very interesting and important. 

Abstract/Introduction

These chapters provide an important introduction to the situation of fires in Indonesia. Nevertheless, they should also provide clear conceptualisation, and I feel there is some overlap in how fires "contribute" to vulnerabilities - sometimes you include "livelihoods" and sometimes not. I assume that the livelihood approach consists of 5 "capitals" covering your focus areas. Moreover, this approach would fit perfectly with your descriptive study (as a framework).

Methodology

The sampling method seems to be clear but lacks references to existing studies and a clear definition of the extent to which it is random/non-random/purposive, etc. Also, I suggest to merge sub-chapters 2.2. with 2.3. for better flow of the text, and similarly 2.4. with 2.5.

Social vulnerability - I miss some memberships or external linkages that provide social back-up in times of crisis.

Economic vulnerability - The shadow economy (subsistence) might influence the final results a lot. An explanation in the chapter Limitation is needed. How was income calculated? As financial income?

Livelihood - It seems to be a mixture of livelihood strategies and demographic/HH head characteristics.

Ecological - Natural resources.

Climate change - First time mentioned.

I think we need some conceptual framework here to keep the consistency of the paper and methodology.

Family - I suggest using "household" as the main economic (task-oriented, incl. vulnerability) unit.

Results

I have a feeling that too many tables make a reader lost. These could be moved to Supplementary files and in the results should be placed more analytical (even from qualitative research) findings.

Also, synthesis with existing sources could be useful. For example, consider merging Results with Discussion.

Discussion

The chapter lacks some coping strategies or alternative strategies. What households do or should do.

Overall layout

The manuscript needs some final tuning to remove all typos, missing values and grammar.

Reviewer 2 Report

Comments and Suggestions for Authors   The title"Farm Household Vulnerability due to Land and Forest Fire in Peatland Areas in South Sumatra" is very interesting and scientific significant. But the content of this manuscript is lack of scientific basis and can not reflect the title , it should be well revised . And what are the indicators for vulnerability classification based on? Are three hundred household samples enough to conclude? The authors should think deeply to discuss these questions. Comments on the Quality of English Language

The manuscript should be checked by the authors carefully and further edited . 

Reviewer 3 Report

Comments and Suggestions for Authors

The article's format is well-organized; however, there are shortcomings in the presentation that require attention.

1. Image 1 is virtually invisible.

2. Throughout the document, various tables are presented, yet they are not referenced in the text.

3. In section 2.5.5. on climate change vulnerability, Table 6 is mentioned. However, the location of this table is not provided.

Lastly, a series of policy proposals are made. How effective do you think they could be? Do you truly believe these measures can be implemented?

From my point of view, it would be better if you present the different indicators in the form of a list, as it is repeated several times (1), (2), etc...

This is just a suggestion, but for example, in section 2.5.2. Economic vulnerability, (1), (2) appears 2 or 3 times without referring to the same thing. As an idea, I propose that you create a table, diagram, or chart with all the indicators for each vulnerability measurement, even if you explain them in the text later. I believe it will be clearer and all the data and/or values used will be gathered.

Round 2

Reviewer 2 Report

Comments and Suggestions for Authors

The manuscript has been well revised and meets the requirements of the journal. It is recommended to publish it.